# Clinical Inference of Serum and Bone Sclerostin Levels in Patients with End-Stage Kidney Disease

**DOI:** 10.3390/jcm8122027

**Published:** 2019-11-20

**Authors:** Annelies De Maré, Anja Verhulst, Etienne Cavalier, Pierre Delanaye, Geert J. Behets, Bjorn Meijers, Dirk Kuypers, Patrick C. D’Haese, Pieter Evenepoel

**Affiliations:** 1Laboratory of Pathophysiology, Department of Biomedical Sciences, University of Antwerp, Universiteitsplein 1, 2610 Wilrijk, Belgium; annelies.demare@uantwerpen.be (A.D.M.); anja.verhulst@uantwerpen.be (A.V.);; 2Department of Clinical Chemistry, University of Liège, Domaine du Sart Tilman, 4000 Liège, Belgium; etienne.cavalier@chuliege.be; 3Department of Nephrology, Dialysis, Hypertension, Transplantation, University of Liège, Domaine du Sart Tilman, 4000 Liège, Belgium; pierre_delanaye@yahoo.fr; 4Nephrology, Division of Internal Medicine, University Hospitals Leuven, Herestraat 49, 3000 Leuven, Belgium; Bjorn.meijers@uzleuven.be (B.M.); Dirk.kuypers@uzleuven.be (D.K.); Pieter.evenepoel@uzleuven.be (P.E.); 5Laboratory of Nephrology, Department of Immunology and Microbiology, KU Leuven, Oude Markt 13, 3000 Leuven, Belgium

**Keywords:** sclerostin, end-stage kidney disease, bone turnover, circulating sclerostin

## Abstract

Mounting evidence indicates that sclerostin, a well-known inhibitor of bone formation, may qualify as a clinically relevant biomarker of chronic kidney disease-related mineral and bone disorder (CKD-MBD), including abnormal mineral and bone metabolism and extraskeletal calcification. For this purpose, in this study we investigate the extent to which circulating sclerostin, skeletal sclerostin expression, bone histomorphometric parameters, and serum markers of bone metabolism associate with each other. Bone biopsies and serum samples were collected in a cohort of 68 end-stage kidney disease (ESKD) patients. Serum sclerostin levels were measured using 4 different commercially available assays. Skeletal sclerostin expression was evaluated on immunohistochemically stained bone sections. Quantitative bone histomorphometry was performed on Goldner stained tissue sections. Different serum markers of bone metabolism were analyzed using in-house techniques or commercially available assays. Despite large inter-assay differences for circulating sclerostin, results obtained with the 4 assays under study closely correlated with each other, whilst moderate significant correlations with skeletal sclerostin expression were also found. Both skeletal and circulating sclerostin negatively correlated with histomorphometric bone and serum parameters reflecting bone formation and turnover. In this study, the unique combined evaluation of bone sclerostin expression, bone histomorphometry, bone biomarkers, and serum sclerostin levels, as assessed by 4 different assays, demonstrated that sclerostin may qualify as a clinically relevant marker of disturbed bone metabolism in ESKD patients.

## 1. Introduction

Sclerostin is a secreted glycoprotein that is encoded by the *SOST* gene [1,2]. It is mainly expressed by the osteocytes, however, other cell types such as chondrocytes have also been shown to produce sclerostin [3]. By binding to its osteoblastic receptor complex, consisting of the low-density lipoprotein receptor-related proteins 5 and 6 (LRP5/6) and the Frizzled (Fz) co-receptors, sclerostin inhibits the Wnt/catenin signaling cascade [4]. As a consequence, bone formation by the osteoblasts is impeded. At the same time, bone resorption is stimulated, since sclerostin induces receptor activator of nuclear factor kappa-Β ligand (RANKL) production by the osteocytes, which in turn induces osteoclastogenesis [5]. 

Mounting evidence indicates that circulating sclerostin may qualify as a biomarker of chronic kidney disease mineral and bone disorder (CKD-MBD) [6]. This biomarker research is greatly hampered by analytical variability. Indeed, according to a recent study, absolute serum sclerostin levels reported for the general, CKD, and dialysis populations largely depend on the assay used [7]. Furthermore, a crucial question remains as to what extent circulating sclerostin levels reflect skeletal sclerostin expression. To clarify this issue, we quantified skeletal sclerostin expression, and for the first time correlated it with circulating sclerostin levels, as determined by 4 commercially available assays. Furthermore, we investigated correlations of skeletal and circulating sclerostin with bone histomorphometric parameters and serum biomarkers of bone formation and turnover. 

## 2. Experimental Section

### 2.1. Study Population

The study population consisted of 68 patients (male, *n* = 19) with end-stage kidney disease (ESKD), recruited from an ongoing prospective observational study at the University Hospital Leuven, Belgium (NCT00547040, NCT01886950). Sixty-four (64) out of the 68 patients were treated with dialysis. The patients were enrolled between September 2010 and July 2013. All participants were 18 years of age or older and provided written informed consent. Serum and bone biopsy were obtained at the time of transplantation. All studies were performed according to the Declaration of Helsinki and approved by the Ethics Committees of the University Hospital Leuven. 

### 2.2. Serum Biochemistry

Following standard centrifugation, serum was aliquoted and stored at −80 °C pending further analysis. Serum sclerostin was measured using four different immunoassays according to the manufacturer’s instructions: DiaSorin LIAISON^®^ chemiluminescent sclerostin assay (Saluggia, Italy), Tecomedical sclerostin high sensitivity enzyme-linked immunosorbent assay (ELISA) kit (TE1023-HS, Sissach, Switzerland), BioMedica human sclerostin ELISA kit (BI-20492, Vienna, Austria), and R&D human SOST (gene encoding for sclerostin) Quantikine ELISA kit (DSST00, Abingdon, UK).

Creatinine, C-reactive protein (CRP), and total calcium and phosphate were measured using standard laboratory techniques. Full-length (bio-intact) parathyroid hormone (PTH) was determined by an immunoradiometric assay, as described elsewhere [8]. Intact fibroblast growth factor 23 (FGF23) (Kainos Laboratories, Tokyo, Japan; reference range (RR): 8–78 pg/mL), osteoprotegerin (OPG) (BioMedica, Vienna, Austria; median concentration (p50) of a healthy population: 2.7 pmol/L), and soluble receptor activator of nuclear factor kappa-Β ligand (sRANKL, BioMedica, Vienna, Austria; p50 of a healthy population: 0.14 pmol/L) were measured according to the manufacturer’s instructions. Interleukin 6 (IL-6) was measured on a Meso QuickPlex SQ120 multiplex imager (Meso Scale Discovery, Rockville, MD, USA) using an electrochemiluminescence multiplex immunoassay (Human Proinflammatory I-4plex, Meso Scale Discovery, Rockville, MD, USA) according to the manufacturer’s instructions. Bone-specific alkaline phosphatase (BsAP; RR: 7.9–25.5 µg/L in men; 6.1–22.2 µg/L and 7.1–23.9 µg/L in pre- and post-menopausal women, respectively), trimeric (“intact”) N-terminal propeptide of type I procollagen (P1NP; RR: 12.8–71.9 µg/L in men, 13.7–71.1 µg/L and < 82.6 µg/L in pre- and post-menopausal women, respectively) and tartrate-resistant acid phosphatase isoform 5b (TRAP5b; RR: 1.4–6.1 U/L in men; 1.2–4.8 U/L and 1.1–6.9 U/L in pre- and post-menopausal women, respectively) were measured with the IDS iSYS instrument (Immunodiagnostic Systems, Boldon, UK). These cut-offs are obviously method-dependent, since large inter-method variation has been observed in CKD patients [9].

Coefficients of variation of all the assays under study were < 10%.

### 2.3. Bone Biopsy and Bone Histomorphometry

A transiliac bone biopsy was performed in 68 ESKD patients at the time of transplantation (general anesthesia), using a needle with an internal diameter of 4.5 mm (Osteobell and Biopsybell, Mirandola, Italy) at a site 2 cm posterior and 2 cm inferior to the anterior iliac spine. 

The method for quantitative histomorphometry of bone has been described in detail elsewhere [10]. Briefly, biopsy specimens were fixed in 70% ethanol and embedded in a methylmethacrylate resin. Undecalcified 5-μm thick sections were stained by the method used by Goldner for quantitative histology to determine static bone parameters. All results are reported as measurements in two dimensions using nomenclature established by the American Society for Bone and Mineral Research [11]. Bone analysis was performed in the Laboratory of Pathophysiology of the University of Antwerp, Belgium, running a custom program on a semi-automatic image analysis system (AxioVision version 4.51, Zeiss Microscopy, Jena, Germany). 

### 2.4. Bone Sclerostin Measurement

Sclerostin expression was investigated on bone sections in order to identify the number of sclerostin-positive osteocytes per bone area. After deacrylation and decalcification, the 4-µm-thick bone sections were blocked with normal goat serum (20% in phosphate buffered saline, PBS) for 20 min and incubated overnight with polyclonal rabbit anti-sclerostin (1:750, ab-63097, Abcam, Cambridge, UK). Biotinylated goat anti-rabbit (Vector Laboratories, California, CA, USA) was used as a secondary antibody. Avidin/biotinylated peroxidase complex (VECTASTAIN ABC kit, Vector Laboratories) was added as signal amplifier and 3-amino-9-ethylcarbazole (AEC, Sigma-Aldrich, Missouri, MO, USA) was used as substrate. The sections were counterstained with hematoxylin. Sections in which the primary antibody was omitted were used as negative controls. The stained bone sections were examined blindly, and sclerostin-positive and sclerostin-negative osteocytic lacunae were counted using the ImageJ manual cell counter plugin (ImageJ 1.8.0, U.S. National Institutes of Health, Bethesda, MD, USA). In order to determine the total bone area (sum of mineralized and not-yet-mineralized bone area), the AxioVision image version 4.51 analysis program was used (Zeiss Microscopy, Jena, Germany). Skeletal sclerostin levels were expressed as number of positive osteocytic lacunae/bone area (µm^2^) and number of positive osteocytic lacunae/number of osteocytic lacunae (percentage positive osteocytic lacunae).

### 2.5. Statistics

Data were analyzed using Prism software (GraphPad Prism 6.0, San Diego, California, CA, USA). Normality was assessed by the Shapiro–Wilk test. Normally distributed data were expressed as mean ± standard deviation (SD). Non-normally distributed data were expressed as the median with interquartile range (IQR). The correlation between the serum and osseous sclerostin concentration, as well as correlations between serum or skeletal sclerostin and demographic data, histomorphometric, and serum parameters of bone metabolism were evaluated by Spearman rank correlation. To test whether the proportion of males and females was equal in high and low skeletal sclerostin expression groups, a chi-square test was performed. A Mann–Whitney U test was performed to compare the level of different parameters between the high and low skeletal sclerostin expression groups, and between genders. A *p*-value < 0.05 was considered to represent a statistically significant difference. 

## 3. Results

### 3.1. Skeletal Sclerostin Expression as a Marker of Bone Turnover

Bone sections of the ESKD cohort were stained for sclerostin expression. The number of sclerostin-positive osteocytic lacunae per total bone area (mineralized area + osteoid area) were counted as a measure of skeletal sclerostin production. Overall, 43% ± 13% (median 43%) of the osteocytic lacunae were positive for sclerostin expression. Expressed as absolute number per µm^2^ bone area, 2.07 ± 0.98 sclerostin-positive osteocytic lacunae were present. 

Histomorphometric and serum markers of bone turnover of the ESKD cohort are presented in Table 1. Results indicate that after dividing the cohort into two groups according to the percentage of sclerostin-positive lacunae (i.e., below versus equal or above the median of 43%) patients with a higher percentage of sclerostin-positive osteocytic lacunae had a significantly lower osteoid area, osteoid perimeter, osteoid width and serum level of BsAP, all markers related to bone formation. In this study population, the prevalence of diabetes (24% of the patients in the low and 16% of the patients in the high skeletal sclerostin expression group) was not significantly different (*p* = 0.4052) between both groups. Furthermore, dividing the cohort in a diabetic and non-diabetic group, did not reveal any significant differences regarding bone (histomorphometric and serum) parameters (Appendix A).

This was confirmed by the correlation matrix in Table 2 showing strong correlations between skeletal sclerostin expression and serum markers of bone metabolism. Mainly osteoid-related parameters (osteoid area, -perimeter, -width, and osteoblast perimeter) were negatively correlated with sclerostin expression at the level of the bone. Also, serum BsAP was significantly (negatively) correlated with skeletal sclerostin expression. Correlations between skeletal sclerostin expression and markers related to bone formation are presented in Figure 1.

### 3.2. Serum Sclerostin Level as a Marker of Skeletal Sclerostin Expression and Bone Turnover

Serum sclerostin levels were determined by four different assays (Figure 2). Median serum sclerostin concentrations showed large differences between the assays. The highest values were obtained using the BioMedica kit (3109 pg/mL, interquartile range (IQR): 2524 pg/mL), whilst the R&D kit detected the lowest values (213 pg/mL, IQR: 159 pg/mL). Serum sclerostin concentrations determined with the 4 assays highly correlated with each other reciprocally (Table 3). 

For the 4 different assays, it was investigated whether serum sclerostin levels was associated with skeletal osteocytic sclerostin expression. Therefore, the cohort again was divided into two groups according to the percentage of sclerostin-positive lacunae (i.e., below the median percentage of 43% and equal or above 43%). Patients with a higher percentage of sclerostin-positive osteocytic lacunae had significantly higher serum sclerostin levels, which is true for all four assays (Table 4). 

This was confirmed by the positive correlation between the % of sclerostin-positive osteocytic lacunae and the serum sclerostin level, which again was true for the four different assays under study. The strength of the correlations was comparable for the four assays under study, as can be seen in Table 5. Skeletal sclerostin explained between 12%–16% of the variability of circulating sclerostin. Similar results were obtained when correlating the serum sclerostin levels with the absolute number of sclerostin-positive osteocytic lacunae per µm^2^ bone area (Appendix A).

Next, it was investigated whether serum sclerostin was related to a series of bone parameters reflecting bone turnover. Bone histomorphometric data presented in Table 6 show that mainly the osteoid-related parameters are negatively correlated with the serum sclerostin levels. Serum sclerostin concentrations, as measured with 3 out of the 4 immunoassays (Tecomedical, BioMedica, and R&D), negatively correlated with both histomorphometric parameters of bone formation (Table 6; osteoblast perimeter, osteoid width, osteoid area, osteoid perimeter) and serum parameters of bone formation (Table 6; BsAP, P1NP), as well as lnPTH (natural logarithmic (ln) transformation of the PTH data). TRAP5b, a bone resorption marker, was also negatively correlated with serum sclerostin levels when measured using the BioMedica assay. Remarkably, the serum sclerostin concentrations measured by the DiaSorin LIAISON assay did not show any significant correlation with these bone turnover markers. Serum sclerostin also showed a positive correlation with OPG and FGF23. 

### 3.3. Correlation between Sclerostin (Serum and Skeletal) and Demographic/Biochemical Variables in the ESKD Cohort

Demographic and biochemical characteristics of the ESKD cohort are presented in Table 7. These characteristics did not differ between the high and low skeletal sclerostin expression groups. Furthermore, associations between these demographic and biochemical serum parameters of the ESKD cohort and skeletal or circulating sclerostin were evaluated by the Spearman rank correlation test (Table 8 and Table 9). In general, the four assays generated comparable correlation coefficients. Serum sclerostin correlated positively with age and BMI, and negatively with residual renal function. Finally, the effect of gender on (skeletal and circulating) sclerostin levels was evaluated, showing that skeletal sclerostin (% sclerostin-positive osteocytic lacunae) was significantly higher in men compared to women, however, there was no (significant) difference in serum sclerostin levels between genders. 

## 4. Discussion

The main findings of the present bone biopsy-based study are: (i) that skeletal sclerostin expression moderately correlates with circulating sclerostin, whatever commercial assay is used; and (ii) that skeletal and circulating sclerostin negatively correlate with histomorphometric and circulating parameters of bone formation. 

Serum sclerostin concentrations determined with the 4 assays highly correlated with each other reciprocally. When considering the absolute values, however, the median serum sclerostin values measured by the four assays under study markedly differed between each other. The highest values (3109 (2524) pg/mL) were found using the BioMedica assay, whilst the lowest values (214 (159) pg/mL) were found using the R&D assay. This is an important finding which should be taken into account when data from different studies using different assays are to be interpreted. 

Serum sclerostin levels measured with the DiaSorin assay did not correlate with the histomorphometric bone parameters, nor with bone formation/resorption biomarkers, contrary to the other assays that found serum sclerostin values to be inversely correlated with bone histomorphometric parameters that are linked to osteoblastic activity and osteoid deposition, hence reflecting bone formation. This is further confirmed by the negative correlation with bone formation markers BsAP and P1NP. 

Despite differences in absolute values for serum sclerostin levels measured with the 4 different assays, results obtained with the 4 assays significantly correlated with the skeletal sclerostin expression. Similar to serum sclerostin levels, skeletal sclerostin expression inversely correlated with bone formation, as evidenced by both histomorphometric and serum markers.

Furthermore, serum sclerostin levels, as measured with the 4 assays under study, were all positively associated with age, BMI, and serum OPG levels. Interestingly, a positive correlation with plasma FGF23 levels was also found with 3 of the 4 assays. 

### 4.1. Interassay Variability

Previous studies [7,12,13,14] have shown that different bioassays for measuring circulating sclerostin show important inter-assay variability. In the present study, we compared the clinical relevance of 4 commercially available sclerostin assays. Inter-assay variability can reasonably be explained by the extent at which interfering substances that have a similar structure to sclerostin or sclerostin fragments are co-detected with the intact molecule. It has been reported that the BioMedica assay detects sclerostin fragments in addition to the intact sclerostin [15]. Furthermore, it has been shown that this assay may also cross-react with proteins that have a similar structure to sclerostin, which was evidenced by the detection of low “sclerostin” protein levels in the serum of sclerosteosis patients (i.e., patients in which functional or intact sclerostin is not expected to be present because of a genetic mutation in the *SOST* gene) [15]. The limited analytical specificity is in line with the fact that with this assay the highest serum sclerostin levels were measured in the current study population. 

According to the manufacturer, the DiaSorin assay was designed to reduce the sources of variation by unbinding heparin and bone morphogenetic proteins (BMP) from sclerostin, in order to make sure that the epitope is available for conjugate recognition [13]. Furthermore, by using antibodies directed against the C- and N-terminus of the sclerostin protein, only intact sclerostin is measured. Given the fact that the Tecomedical assay also results in higher sclerostin concentrations compared to the DiaSorin assay, this might imply that the former assay also detects fragments or cross-reacts with proteins that are structurally similar to sclerostin. 

Lastly, the R&D assay systematically gave the lowest sclerostin values, which suggests that only intact sclerostin is measured [14]. Yet, there are large differences in measured serum sclerostin levels between the DiaSorin and the R&D assay. Some of the variation could be due to the detection of dimeric or other sclerostin-associated protein complexes with the DiaSorin assay as opposed to the R&D assay.

### 4.2. Circulating vs. Bone Sclerostin

Importantly, despite the differences in absolute sclerostin values obtained with the different immunoassays, comparable correlations were found with the % of sclerostin-positive osteocytic lacunae (i.e., a measure of skeletal protein expression). These correlations, although significant, were rather modest. To a certain extent, this might be explained by limitations specific to the assays or the quantification of skeletal sclerostin expression. Moreover, examining sclerostin expression in bone biopsies does not necessarily reflect the whole skeletal compartment. Indeed, skeletal heterogeneity of sclerostin expression is probable, in part reflecting differences in mechanical loading between different skeletal sites [16]. In addition, extra-osseous-produced sclerostin has been hypothesized to spill over into the circulation, and as such represents an additional confounding factor (source of bias) [17]. In this regard, especially the vasculature needs to be mentioned. During the vascular calcification process, vascular smooth muscle cells undergo osteochondrogenic transdifferentiation, which goes along with increased activation of the Wnt/β-catenin signaling cascade [18]. Although debate is still ongoing, experimental as well as clinical studies have demonstrated increased expression of sclerostin in calcifying vascular smooth muscle cells and in aortic valves [17,19,20,21]. This has led to the interesting hypothesis that sclerostin produced in the calcified vessels might partly spill-over to the serum compartment, leading to increased serum sclerostin levels in patients or animals with vascular media calcification [22,23]. The relationship between serum sclerostin, vascular sclerostin expression, and vascular calcification, however, requires further investigation. 

### 4.3. Demographic and Biochemical (Serum) Parameters in the ESKD Cohort in Relation to Sclerostin

This study confirmed a negative association between serum sclerostin and residual renal function [24]. While originally explained as the consequence of renal retention [25,26], Cejka et al., provided evidence that the increasing serum sclerostin levels in patients with declining kidney function most likely results from an increased (possibly extra-osseous) production [27]. On the other hand, one might also argue that the lower the RRF is, the higher the concentration of circulating sclerostin fragments. Hence, with less specific assays measuring both the intact form and fragments, higher concentrations will reasonably be found. This latter assumption is supported by our observation that with the R&D assay, which yielded the lowest mean serum sclerostin levels, no correlation was found with RRF, in contrast to the other assays under study.

The increased serum sclerostin levels in men versus women were not confirmed in the present study [25,26]. This is probably due to the unequal distribution between men (*N* = 19) and women (*N* = 49). It should be mentioned, however, that a significant positive correlation between skeletal sclerostin expression and male gender was observed.

As also reported by others [28,29], the BMI of our cohort was positively correlated with serum sclerostin levels. Currently, no clear explanation is available. One hypothesis could be based on the observation that sclerostin production is increased in individuals that are less active compared to the physically active individuals [28]. During physical activity, mechanical stress is applied to the skeleton, which is sensed by the osteocytes, which then respond by lowering sclerostin expression [16]. Another hypothesis is a possible role for sclerostin in adipogenesis [23,30,31,32]. Research showed that mice overexpressing the *Sost* gene (encoding for sclerostin) had excess adipose tissue [31]. 

Three out of four assays also confirmed the previously reported positive correlation between plasma FGF23 levels and serum sclerostin levels in CKD patients [29,33], which as evidenced from experimental studies is due to the fact that sclerostin inhibits the protein-encoding gene PHEX [34]. PHEX reduces FGF23 activity by interacting with its co-receptor Klotho [33]. Inhibition of PHEX by sclerostin indirectly results in an increased tubular phosphate excretion and decreased 1,25 (OH)_2_ Vit D synthesis via unhindered FGF23 activity. These findings are in line with data from a study demonstrating sclerostin knockout mice to have decreased FGF23 concentrations, which results in increased serum phosphate and 1,25 (OH)_2_ Vit D levels [34]. Our results are in line with the physiologic regulatory role of sclerostin on FGF23 synthesis, however, alternative explanations cannot be excluded, such as concomitant confounding by residual renal function. 

Furthermore, in our ESKD cohort serum sclerostin strongly correlated with serum OPG levels in a positive way. Apparently, this is rather unexpected, since sclerostin, a protein with catabolic effects on bone, is known to inhibit osteoblastic OPG production [5]. As a decoy receptor of RANKL, OPG inhibits bone resorption, and mice lacking OPG have been shown to suffer from osteoporosis [35]. On the other hand, OPG levels are known to be increased in patients with vascular calcification [36], and mice lacking OPG develop vascular media calcification in addition to osteoporosis [35]. Furthermore, OPG is identified as a vascular smooth muscle cell-specific senescence-associated secretory phenotype (SASP) protein [37], and therefore, similar to sclerostin, serum OPG levels in vascular calcification might increase as a result of extra-osseous (vascular) OPG production.

As demonstrated by others [12,38,39,40,41,42], an inverse association was observed between serum sclerostin and serum lnPTH concentrations. PTH is known to be a regulator of sclerostin production by reducing the sclerostin expression. Remarkably, serum sclerostin levels measured by the DiaSorin assay, as opposed to the other assays, did not significantly correlate with serum lnPTH levels, nor did they correlate with histomorphometrical or serum parameters related to bone formation.

### 4.4. Sclerostin and Bone Turnover

In the ESKD cohort under study, both serum sclerostin (exception made for the DiaSorin assay) and skeletal sclerostin levels inversely correlate with both histomorphometric and serum markers of bone formation. This indicates that the serum sclerostin concentration (again an exception is made for the DiaSorin assay) can be considered a potential marker of bone turnover in ESKD patients. The fact that serum sclerostin and bone turnover negatively correlate aligns with the current paradigm that sclerostin inhibits bone formation. 

## 5. Conclusions

Serum sclerostin levels measured by different assays correlate well with each other, however, significantly differ in terms of absolute levels, which implies that data from different studies using assays from different manufacturers should be interpreted with caution. 

In ESKD patients, regardless of the commercial assay used, circulating sclerostin significantly correlates with skeletal sclerostin and histological or biochemical parameters of bone formation. 

Overall, circulating sclerostin levels may be considered a relevant biomarker of disturbed bone metabolism in ESKD patients. 

## Figures and Tables

**Figure 1 jcm-08-02027-f001:**
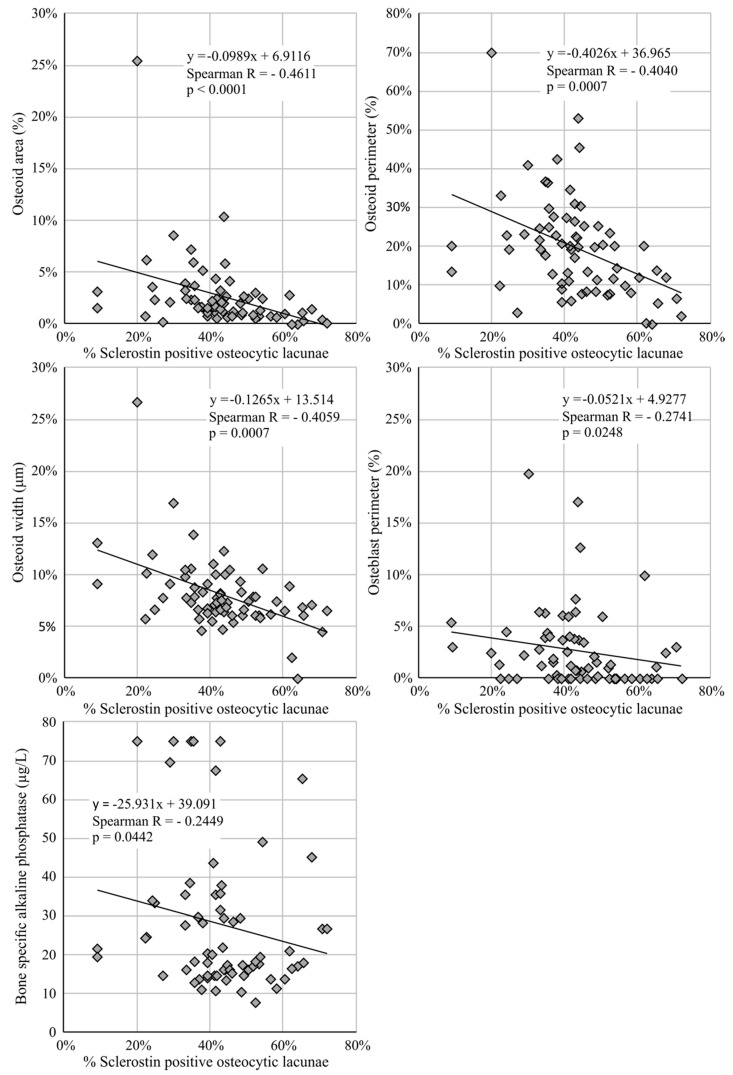
Negative correlation between the percentage of sclerostin-positive osteocytic lacunae and the osteoid area, osteoid perimeter, osteoid width, osteoblast perimeter, and bone-specific alkaline phosphatase.

**Figure 2 jcm-08-02027-f002:**
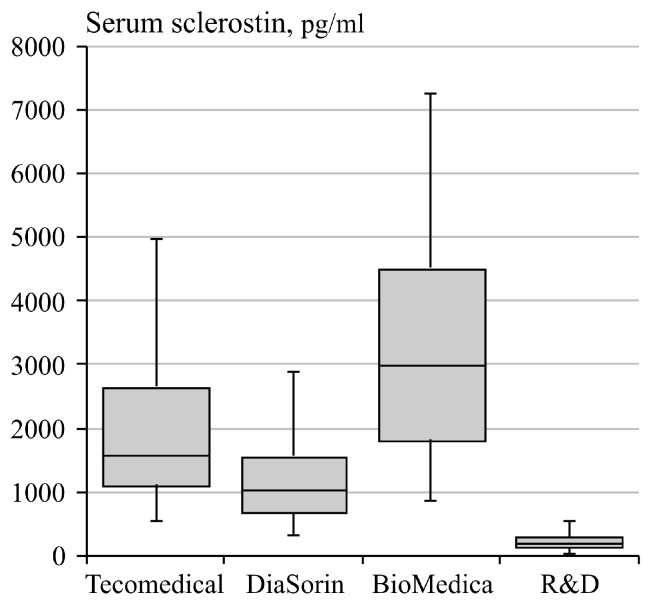
Serum sclerostin levels in the end-stage kidney disease (ESKD) cohort, measured using four different commercially available kits (median, 1st and 3rd percentiles, minimum and maximum).

**Table 1 jcm-08-02027-t001:** Bone histomorphometric parameters and serum markers of bone metabolism of the total end-stage kidney disease (ESKD) cohort and according to the percentage of sclerostin-positive lacunae (i.e., below versus equal or above the median of 43%).

	ESKD Patients All		Sclerostin-Positive Lacunae < 43%		Sclerostin-Positive Lacunae ≥ 43%		*p*-Value
Histomorphometric bone parameters							
B.Ar (%)	22.11	±6.44	22.35	±6.49	21.83	±6.48	0.5239
Min.Area (%)	98.3	(2.10)	97.61	(2.42)	98.60	(1.77)	0.0166 *
O.Ar (%)	1.70	(2.09)	2.31	(2.07)	1.14	(1.81)	0.0041 **
O.Pm (%)	19.92	(14.95)	21.82	(15.31)	12.89	(12.95)	0.0124 *
E.Pm (%)	4.42	(4.92)	4.41	(4.12)	4.65	(5.67)	0.7861
O.Wi (m)	7.408	(2.768)	7.863	(3.369)	7.154	±2.055	0.0144 *
Ob.Pm(T) (%)	1.33	(3.98)	2.46	(4.47)	0.72	(2.61)	0.0512
Oc.Pm(T) (%)	0.63	(1.17)	0.57	(1.18)	0.65	(1.25)	0.8449
Tb.Th (m)	145.4	(42.0)	144.7	±26.0	143.3	(45.5)	0.7187
Tb.N (mm^−1^)	1.936	±0.4654	2.001	(0.5190)	1.888	±0.4586	0.4019
Tb.Sp (m)	379.7	(188.0)	372.9	(161.7)	414.6	±142.6	0.3296
Serum markers of bone metabolism							
BsAP (g/L)	19.95	(18.48)	27.60	(22.80)	17.50	(11.40)	0.0152 *
P1NP (g/L)	83.65	(72.17)	91.00	(81.70)	72.20	(51.00)	0.1402
TRAP5b (U/L)	5.74	(3.63)	6.45	±2.63	5.32	±2.23	0.0641
lnPTH (pg/mL)	5.3	(1.1)	5.40	(0.70)	5.00	(1.30)	0.0577
FGF23 (pg/mL)	1434.0	(6806.8)	1516.0	(6786.9)	1129.0	(6990.7)	0.8998
OPG (ng/mL)	9.2	(6.8)	8.4	(2.3)	10.64	(6.38)	0.4034
sRANKL (ng/mL)	0.087	(0.098)	0.086	(0.101)	0.090	(0.095)	0.8366

Values are presented as mean ± SD or median (IQR) when variables were not normally distributed (*n* = 68). Note: * *p* ≤ 0.05, ** *p* ≤ 0.01. Abbreviations: B.Ar = bone area (% of tissue area); Min.Ar = mineralized bone area (% of bone area); O.Ar = osteoid area (% of bone area); O.Pm = osteoid perimeter; E.Pm = eroded perimeter; O.Wi = osteoid width; Ob.Pm(T) = osteoblast perimeter (relative to the total perimeter); Oc.Pm(T) = osteoclast perimeter (relative to the total perimeter); Tb.Th = trabecular thickness; Tb.N = trabecular number; Tb.Sp = trabecular spacing; BsAP = bone-specific alkaline phosphatase; P1NP = procollagen type 1 N-terminal propeptide; TRAP5b = tartrate-resistant acid phosphatase 5b; lnPTH = natural logarithm parathyroid hormone; FGF23 = fibroblast growth factor 23; OPG = osteoprotegerin; sRANKL = soluble receptor activator of nuclear factor kappa-B ligand.

**Table 2 jcm-08-02027-t002:** Spearman correlation matrix of skeletal sclerostin expression vs. histomorphometric bone parameters and serum markers of bone metabolism.

	% Sclerostin-Positive Osteocytic Lacunae	
	Spearman *r*	*p*-Value
**Histomorphometric bone parameters**		
**B.Ar**	−0.1432	0.2476
**Min.Area**	0.3259	0.0071 **
**O.Ar**	−0.4611	<0.0001 ****
**O.Pm**	−0.4044	0.0007 ***
**E.Pm**	−0.0023	0.9857
**O.Wi**	−0.4059	0.0007 ***
**Ob.Pm(T)**	−0.2741	0.0248 *
**Oc.Pm(T)**	0.0798	0.5210
**Tb.Th**	0.0666	0.5926
**Tb.N**	−0.1577	0.2024
**Tb.Sp**	0.1777	0.1503
**Serum markers of bone metabolism**		
**BsAP**	−0.2449	0.0442 *
**P1NP**	−0.1745	0.1547
**TRAP5b**	−0.2028	0.0972
**lnPTH**	−0.1535	0.2185
**FGF23**	0.0619	0.6160
**OPG**	0.1664	0.1750
**sRANKL**	0.0214	0.8623

Note: * *p* ≤ 0.05, ** *p* ≤ 0.01, *** *p* ≤ 0.001, **** *p* ≤ 0.0001. Spearman r: Spearman’s rank correlation coefficient.

**Table 3 jcm-08-02027-t003:** Spearman correlation matrix of serum sclerostin concentration, measured by 4 different assays. Note: **** *p*-value < 0.0001.

	Tecomedical	DiaSorin	BioMedica	R&D
**Tecomedical**		0.902 ****	0.915 ****	0.826 ****
**DiaSorin**	0.902 ****		0.878 ****	0.811 ****
**BioMedica**	0.915 ****	0.878 ****		0.880 ****
**R&D**	0.826 ****	0.811 ****	0.880 ****	

**Table 4 jcm-08-02027-t004:** Serum sclerostin levels of the total end-stage kidney disease (ESKD) cohort and according to the percentage of sclerostin-positive lacunae (i.e., below versus equal or above the median of 43%).

	ESKD Patients All		Sclerostin-Positive Lacunae < 43%		Sclerostin-Positive Lacunae ≥ 43%		*p*-Value
**Serum sclerostin levels (pg/mL)**							
**Tecomedical**	1687	(1501)	1524	(998)	2122	(1642)	0.0067 **
**DiaSorin**	1155	(848)	1010	(645)	1350	(1307)	0.0328 *
**BioMedica**	3109	(2524)	2512	(2374)	4075	±1708	0.0036 **
**R&D**	214	(159)	164	(165)	284	±135	0.0051 **

Note: * *p* ≤ 0.05, ** *p* ≤ 0.01.

**Table 5 jcm-08-02027-t005:** Spearman correlation matrix of skeletal sclerostin expression vs. serum sclerostin levels.

	% Sclerostin-Positive Osteocytic Lacunae	
	Spearman *r*	*p*-Value
**Serum sclerostin levels**		
**Tecomedical**	0.3720	0.0018 **
**DiaSorin**	0.3457	0.0039 **
**BioMedica**	0.4044	0.0006 ***
**R&D**	0.3752	0.0016 **

Note: ** *p* ≤ 0.01, *** *p* ≤ 0.001.

**Table 6 jcm-08-02027-t006:** Spearman correlation matrix of serum sclerostin concentration vs. histomorphometric bone parameters and serum markers of bone metabolism.

	Tecomedical		DiaSorin		BioMedica		R&D	
**Histomorphometric bone parameters**								
**B.Ar**	−0.0287	0.8175	−0.0437	0.7252	−0.0159	0.8986	−0.0691	0.5786
**Min.Area**	0.1444	0.2435	0.1270	0.3058	0.1930	0.1176	0.1429	0.2487
**O.Ar**	−0.2005	0.1038	−0.1705	0.1678	−0.2605	0.0333 *	−0.2219	0.0711
**O.Pm**	−0.1848	0.1343	−0.1636	0.1858	−0.2594	0.0340 *	−0.2286	0.0628
**E.Pm**	−0.1828	0.1451	−0.1402	0.2655	−0.1873	0.1351	−0.1348	0.2842
**O.Wi**	−0.2577	0.0306 *	−0.2213	0.0719	−0.2508	0.0406 *	−0.3014	0.0132 *
**Ob.Pm(T)**	−0.3013	0.0125 *	−0.1988	0.1068	−0.2785	0.0225 *	−0.2164	0.0786
**Oc.Pm(T)**	−0.1546	0.2115	−0.1677	0.1750	−0.1686	0.1726	−0.2031	0.0993
**Tb.Th**	−0.2008	0.1032	−0.1923	0.1190	−0.1672	0.1763	−0.2635	0.0312 *
**Tb.N**	0.1327	0.2844	0.1077	0.3856	0.1454	0.2405	0.1660	0.1794
**Tb.Sp**	−0.1021	0.4111	−0.0778	0.5317	−0.1150	0.3542	−0.1085	0.3819
**Serum markers of bone metabolism**								
**BsAP**	−0.2727	0.0245 *	−0.1596	0.1935	−0.2950	0.0146 *	−0.2761	0.0227 *
**P1NP**	−0.3182	0.0082 **	−0.1717	0.1616	−0.3655	0.0022 **	−0.3015	0.0125 *
**TRAP5b**	−0.2351	0.0536	−0.1245	0.3118	−0.2656	0.0286 *	−0.1434	0.2434
**lnPTH**	−0.2566	0.0376 *	−0.1694	0.1740	−0.3274	0.0073 **	−0.3571	0.0032 **
**FGF23**	0.2754	0.0230 *	0.3863	0.0011 **	0.2088	0.0874	0.2579	0.0337 *
**OPG**	0.4379	0.0002 ***	0.3823	0.0013 **	0.4108	0.0005 ***	0.4123	0.0005 ***
**sRANKL**	<0.0001	0.9998	0.0602	0.6256	−0.0272	0.8259	−0.0467	0.7053

Note: * *p* ≤ 0.05, ** *p* ≤ 0.01, *** *p* ≤ 0.001.

**Table 7 jcm-08-02027-t007:** Demographic and biochemical characteristics of the total end-stage kidney disease (ESKD) cohort and according to the percentage of sclerostin-positive lacunae (i.e., below versus equal or above the median of 43%).

	ESKD Patients, All		Sclerostin-Positive Lacunae < 43%		Sclerostin-Positive Lacunae ≥ 43%		*p*-Value
Clinical and biochemical characteristics							
Age	57	(21)	58	(25)	55	±10.34	0.8377
BMI	26	±5	25	±5	26	±5	0.2926
Renal residual function (mL/24 h)	500	(963)	707	±672	741	(1000)	0.9508
Dialysis vintage (months)	30.4	±18.1	29.6	±17.2	31.4	±19.3	0.8424
Serum Calcium (mg/dL)	9.4	(1.0)	9.4	(0.9)	9.4	±0.7	0.3625
Serum Phosphate (mg/dL)	4.6	±1.4	4.6	±1.5	4.6	±1.4	0.9626
Serum CRP (mg/L)	3.3	(5.0)	3.6	(5.1)	3.0	(6.7)	0.4871
Serum IL-6 (pg/mL)	1.4	(2.1)	1.6	(2.6)	1.2	(2.1)	0.0819

**Table 8 jcm-08-02027-t008:** Spearman correlation matrix of skeletal sclerostin expression vs. demographic and biochemical characteristics. Note: CRP = C-reactive protein.

	% Sclerostin-Positive Osteocytic Lacunae	
	Spearman *r*	*p*-Value
Clinical and biochemical characteristics		
Age	0.07831	0.5256
BMI	0.2074	0.1619
Renal residual function	−0.08871	0.5156
Serum Calcium	0.1003	0.4194
Serum Phosphate	0.0308	0.8049
Serum CRP	0.1113	0.3737
Serum IL-6	−0.1052	0.3934

**Table 9 jcm-08-02027-t009:** Spearman correlation matrix of serum sclerostin concentration vs. demographic and biochemical characteristics.

	Tecomedical		DiaSorin		BioMedica		R&D	
Clinical and biochemical characteristics								
**Age**	0.3088	0.0104 *	0.3016	0.0124 *	0.3123	0.0095 **	0.2436	0.0453 *
**BMI**	0.3937	0.0062 **	0.3887	0.0069 **	0.5159	0.0002 ***	0.4021	0.0051 *
**RRF**	−0.3814	0.0037 **	−0.5741	<0.0001 ****	−0.2892	0.0306 *	−0.2544	0.0585
**Serum Calcium**	0.0069	0.9572	0.0816	0.5115	0.0825	0.5069	0.1406	0.2565
**Serum Phosphate**	0.0873	0.4825	0.0644	0.6046	−0.0285	0.8191	0.0758	0.5419
**Serum CRP**	0.1855	0.1359	0.1263	0.3123	0.1551	0.2136	0.2049	0.0989
**Serum IL-6**	0.0660	0.5927	0.1312	0.2863	−0.0020	0.9872	−0.0058	0.9624

Note: * *p* ≤ 0.05, ** *p* ≤ 0.01, *** *p* ≤ 0.001, **** *p* ≤ 0.0001.

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
