# Peer review of "Clinical Inference of Serum and Bone Sclerostin Levels in Patients with End-Stage Kidney Disease"

_jcm, 2019, doi:10.3390/jcm8122027_

Round 1
Reviewer 1 Report
The authors concluded that serum sclerostin correlated with histological and biochemical parameters in ESKD patients. As the authors pointed out, sclerostin is a unique biomarker for bone metabolism, and in addition, it might associate with ectopic calcification including arterial wall and cardiac valve. From the above reasons, I considered that the present article was worth publishing. However, there are some points I would like to make sure and recommend to be revised.
1. The authors described the subjects were ESKD patients. Have all the patients been dialyzed? If so, dialysis vintage and dialysis efficacy such as Kt/V should be shown. If not, kidney function should be shown.
2. Serum sclerostin level is reported to be lower in diabetes patients. What percentage did diabetes patients account for between the 2 groups? I would like to know association between comorbidity of diabetes and biomarkers and bone histology.
3. Which assay did the authors consider the best? In addition, how did the authors choose the different assays?
4. I considered that some medication including calcium supplementation, phosphate binders, and activated vitamin D might influence both serum sclerostin and bone histology. Hence, the authors should describe the usage of the medications.
5. The authors indicated that serum sclerostin level measured by the 3 assays was correlated with RRF. I thought that RRF itself was strongly associated with bone biomarker. I would like to know the authors’ opinions.
Author Response
The authors described the subjects were ESKD patients. Have all the patients been dialyzed? If so, dialysis vintage and dialysis efficacy such as Kt/V should be shown. If not, kidney function should be shown.
All patients were ESKD patients (scheduled for renal transplantation); 64 out of the 68 patients were treated with dialysis. We included this information in the manuscript.
Data on Kt/V are unfortunately lacking. Information on dialysis vintage and RRF was already included in the manuscript.
Serum sclerostin level is reported to be lower in diabetes patients. What percentage did diabetes patients account for between the 2 groups? I would like to know association between comorbidity of diabetes and biomarkers and bone histology.
The number of patients with diabetes was not significantly different between both groups (24% in the low and 16% in the high skeletal sclerostin expression group). We added this information to the manuscript.
Although we agree that the effect of DM on bone turnover parameters (histological as well as serum parameters by itself might be an interesting issue, we believe that this is beyond the scope of this paper.
For the reviewers info however, we calculated p-values for possible differences in bone (histomorphometric and serum) parameters, between diabetic and non-diabetic patients. No significant differences were found as indicated by the p-values shown in table below.
|
P-value |
|
|
lnPTH |
0,445 |
|
FGF23 |
0,9701 |
|
TRAP5b |
0,8513 |
|
sRANKL |
0,9368 |
|
OPG |
0,1137 |
|
P1NP |
0,5135 |
|
BsAP |
0,1393 |
|
BAR |
0,909 |
|
OAR |
0,2679 |
|
Mar |
0,7219 |
|
Opm |
0,611 |
|
Epm |
0,9698 |
|
Owi |
0,3249 |
|
ObPmT |
0,351 |
|
OcPmT |
0,4137 |
|
TbTh |
0,2066 |
|
TbN |
0,7086 |
|
TbSp |
0,7086 |
Which assay did the authors consider the best? In addition, how did the authors choose the different assays?
Results obtained with the 4 assays under study strongly correlate with each other. It is therefore difficult to consider one particular assay as being the best one.. In case, one is interested in measuring serum sclerostin concentrations to get insight in bone metabolism, then the Biomedica, R&D and Tecomedical assays should be preferred to the Diasorin assay. This was already mentioned in the Discussion of the manuscript at first submission.
These assays were chosen since they are very often used and were easily available (1-3).
I considered that some medication including calcium supplementation, phosphate binders, and activated vitamin D might influence both serum sclerostin and bone histology. Hence, the authors should describe the usage of the medications.
The primary aim of the present study was to investigate the association between circulating and bone sclerostin. We acknowledge that some medications such as bisphosphonates, denosumab and glucocorticoids may affect skeletal sclerostin expression. At present, we do not have detailed information on medication use. As our patients were all dialysis patients, it is unlikely that they were treated with these medications. Furthermore, it may be anticipated that these medications will not affect the relationship between skeletal and circulating sclerostin. Therefore, we do not readily expect medication to have a significant impact on the general conclusions in this paper. However, if the editor argues different, we’re willing to extract this information from the medical files, but to do so we need additional time.
The authors indicated that serum sclerostin level measured by the 3 assays was correlated with RRF. I thought that RRF itself was strongly associated with bone biomarker. I would like to know the authors’ opinions.
Of all bone biomarkers under study, only lnPTH was significantly correlated with kidney function (see table below).
In addition to its correlation with lnPTH, serum sclerostin levels also significantly correlated with BsAP, P1NP, FGF23 and OPG, which were not correlated with RRF. Data thus indicate that these correlations cannot be attributed to a common dependence on RRF.
|
|
RRF vs |
||||||
|
|
lnPTH |
FGF23 |
TRAP5b |
sRANKL |
OPG |
P1NP |
BsAP |
|
Spearman r |
-0.3629 |
-0.2297 |
-0.0972 |
-0.172 |
-0.1619 |
-0.1117 |
-0.1488 |
|
P-value |
0.007 ** |
0.0886 |
0.4760 |
0.2050 |
0.2331 |
0.4126 |
0.2737 |
Reviewer 2 Report
In this cross-sectional study of 68 patients with end-stage kidney disease, the authors made several important findings; 1) inter-assay variability of serum sclerostin measurements, 2) correlations between circulating and bone sclerostin, and 3) associations between sclerostin levels and histomorphometric bone parameters. This paper provides fundamental and valuable information for readers in this field. I would like to make a few comments:
The authors used “the number of sclerostin positive osteocytic lacunae per total bone areas” as a surrogate of the bone sclerostin expressions. In this method, however, the information about the absolute number of sclerostin-positive osteocytic lacunae is lost. The smaller percentages of sclerostin positive osteocytic lacunae might mean the larger numbers of osteocytic lacunae. How about the relationship between the absolute number of sclerostin-positive osteocytic lacunae and serum sclerostin levels?
The positive correlation between BMI and serum sclerostin concentrations may be confounded by physical activity levels. The authors should comment on this point in the Discussion (4.3.)
Please specify the actual P-values throughout Tables like Table 7.
Author Response
The authors used “the number of sclerostin positive osteocytic lacunae per total bone areas” as a surrogate of the bone sclerostin expressions. In this method, however, the information about the absolute number of sclerostin-positive osteocytic lacunae is lost. The smaller percentages of sclerostin positive osteocytic lacunae might mean the larger numbers of osteocytic lacunae. How about the relationship between the absolute number of sclerostin-positive osteocytic lacunae and serum sclerostin levels?
Although not mentioned in this manuscript, we also correlated serum sclerostin concentrations with the absolute number of sclerostin positive osteocytic lacunae/μm2 bone area).
|
Number of sclerostin positive osteocytic lacunae/µm2 bone area vs serum sclerostin levels: |
Spearman r |
P-value |
|
- Tecomedical |
0.3040 |
0.0117 * |
|
- Diasorin |
0.3301 |
0.0060 ** |
|
- Biomedica |
0.2887 |
0.0170 * |
|
- R&D |
0.2574 |
0.0341 * |
Since stronger correlations were obtained when using ‘% sclerostin positive osteocytic lacunae’, we preferred to show the correlations of with the percentages and did not perform further analysis with the absolute numbers in order not to make the paper too complex. A supplemental file with the data presented in table above has now been submitted also.
The positive correlation between BMI and serum sclerostin concentrations may be confounded by physical activity levels. The authors should comment on this point in the Discussion (4.3.)
Although we can’t confirm, we agree that an increased BMI might have been associated with lower physical activity (which is known to be related to increased sclerostin production)(4, 5). Another hypothesis is the involvement of sclerostin in adipogenesis (6). We have included this in the manuscript.
Please specify the actual P-values throughout Tables like Table 7.
P-values have now been included in response to the reviewer’s suggestion
Round 2
Reviewer 1 Report
The authors have revised approproately.